# Virtual Reality Systems as an Orientation Aid for People Who Are Blind to Acquire New Spatial Information

**DOI:** 10.3390/s22041307

**Published:** 2022-02-09

**Authors:** Orly Lahav

**Affiliations:** The Constantiner School of Education, Tel Aviv University, Tel Aviv 6997801, Israel; lahavo@tauex.tau.ac.il

**Keywords:** blind, virtual reality, orientation and mobility, spatial perception, cognitive map

## Abstract

This research aims to examine the impact of virtual environments interface on the exploration process, construction of cognitive maps, and performance of orientation tasks in real spaces by users who are blind. The study compared interaction with identical spaces using different systems: BlindAid, Virtual Cane, and real space. These two virtual systems include user-interface action commands that convey unique abilities and activities to users who are blind and that operate only in these VR systems and not in real space (e.g., teleporting the user’s avatar or pointing at a virtual object to receive information). This research included 15 participants who are blind, divided into three groups: a control group and two experimental groups. Varied tasks (exploration and orientation) were used in two virtual environments and in real spaces, with both qualitative and quantitative methodologies. The results show that the participants were able to explore, construct a cognitive map, and perform orientation tasks. Participants in both virtual systems used these action commands during their exploration process: all participants used the teleport action command to move their avatar to the starting point and all Virtual Cane participants explored the environment mainly by using the look-around mode, which enabled them to collect spatial information in a way that influenced their ability to construct a cognitive map based on a map model.

## 1. Introduction

People who are blind face deficits in the ability to navigate outdoor and new indoor spaces. The lack of the sense of sight makes it difficult to identify obstacles and locations independently, or simply to find a target path. Consequently, people who are blind must use compensatory multisensorial (touch, audio, and olfactory) channels and alternative exploration methods [1].

Over the past 50 years, a large number of orientation and mobility (O&M) digital technologies have been developed and researched [2]. The increased number of new O&M digital aids for people who are blind has had positive effects on O&M rehabilitation programs and their users. The new digital orientation aids provide spatial information (in advance or in situ) about unexplored spaces.

A virtual reality (VR) system can enhance the capabilities of people with sensorial, physical, mental, and learning disabilities in multiple areas [3,4]. Research and development of orientation VR for people who are blind have been conducted over the past 25 years. A survey [5] that included VR systems for people who are blind and visually impaired over the past two decades clustered the VR systems into a three-level taxonomy based on exploration interaction, perspective, application scenario, and evaluation. These VR systems compensate for lacking visual information through haptic and/or auditory feedback. The haptic feedback transmits sensation through direct interaction with the virtual object (e.g., texture and/or stiffness) to allow detection of artificial representations of real objects. The haptic devices include SensAble Phantom Desktop, Immersion Corp.’s CyberForce, and Novint Falacon, and the tactile devices include the force feedback joystick and Nintendo’s Wii Controller. The auditory stimulus can include mono, stereo, or surrounding audio that allows the user to detect the direction and distance of sounds, which are then used as clues or landmarks. Past research on VR for people who are blind has revealed the benefits of such multisensorial systems. These benefits support people who are blind in perceiving spatial information, spatial problem solving, practicing and enhancing O&M skills, and building O&M strategies [6]; enabling the user’s independent interaction; displaying immediate feedback suiting the user’s sensory and cognitive abilities; and providing the opportunity to practice in a safe area, without time limitations or professional restrictions.

Additionally, orientation virtual environments (VEs) can support O&M specialists in the rehabilitation training process [6]. Most VR systems include indoor and outdoor spaces that allow learners who are blind to explore a new space in advance. While exploring the VE, the learner interacts with the landmarks and clues and collects spatial information that will later support him or her in constructing a cognitive map that can be applied in real space (RS). A few research teams have developed and researched orientation VEs for users who are blind, such as [7,8,9,10,11,12,13,14,15,16,17]. Their research findings showed that people who are blind were able to explore VE systems independently, to construct cognitive maps as a result of the exploration, and to apply this spatial knowledge successfully in familiar and unfamiliar RSs. Other orientation VR systems have been used mainly to help trainees who are blind to acquire spatial and O&M skills [9,18,19,20,21]. These research findings have indicated the potential of VR systems to play a central role in three activities: as an exploration/navigation planning tool for independent traveling in unfamiliar RSs, as a training simulator for orientation, and as a diagnostic tool for O&M specialists to track and observe learners’ spatial abilities and strategies.

The VR system does have several limitations affecting all of the above uses. The VE is not a replica or replacement for a rehabilitation specialist’s instruction or for exploration in RS; however, in cases in which the RS is not accessible for exploration, the VE is a good substitute.

Traditional O&M rehabilitation programs provide practice in the acquisition of O&M skills and spatial mapping, which are supplied at the perceptual and conceptual levels. At the perceptual level, people who are blind are able to collect multisensorial spatial information about their surroundings and apply it to orient themselves in indoor and outdoor spaces [22]. At the conceptual level, the focus is on developing compensatory exploration strategies to perceive a spatial representation that can be applied efficiently in RS. Spatial representation is stored as a cognitive map—a mental representation of an image based on one’s knowledge about a space [23]. The cognitive map can be represented as a route model, in which the space is described in terms of a series of displacements in space; as a map model, in which the space is described from a bird’s-eye view of the space; or as an integrated mental representation of both route and map models. Most O&M rehabilitation programs trainees are directed to use the route model as a spatial model to promote safety and to allow concentration on the target path.

People who are blind principally rely on a route model to construct a cognitive map [24]. They will usually require a map model upon encountering an unusable path (due to a fallen tree, road construction, etc.), engendering the need to identify an alternate route. Furthermore, people who are blind are not usually trained to explore map layers (e.g., surface, routes, traffic, public transportation, or buildings). The development of new VR systems can have value in filling these gaps by including special action commands that are unique to the VE, a benefit that is not available in RS to the independent explorer who is blind. The development of new O&M digital technologies must also take into account the challenges of developing simple and intuitive user interfaces [25], especially user interfaces accessible to people who are blind [26,27].

This study compared two orientation VR systems. Both systems contain user interfaces with unique components that enable the explorer who is blind to work independently. Both convey to the user abilities and activities that operate only in these VR systems and are not available in RS. The two VR systems, the BlindAid system [9,28] and the Virtual Cane (Wiimote) [29], have been previously studied. In this research, we examine and compare the spatial behavior of participants who are blind exploring two multisensorial VR systems. The VEs were identical and represented corresponding simple and complex RSs. Each group of participants explored one VR system or RS. The two main goals of the research were: first, to understand the impact of a multisensorial VE on spatial abilities for users who are blind by comparing their exploration in identical unfamiliar spaces using different VR systems or RS; and second, to examine the use of unique user-interface action commands and their influence on the exploration process, the construction of cognitive maps, and the transfer of this spatial knowledge during O&M in the RS. This study explores three research questions:How do people who are blind explore unfamiliar spaces using BlindAid or Virtual Cane, or in RS?What were the participants’ cognitive mapping characteristics after exploration using the BlindAid or Virtual Cane, or in RS?How did the control group and the two experimental groups (BlindAid and Virtual Cane) perform the RS orientation tasks?

## 2. Materials and Methods

### 2.1. Participants

This research included 15 participants; six criteria were used to choose the research participants: totally blind without residual vision; having onset of blindness at least two years prior to the research period; without other disabling condition; instructed previously in O&M; understanding English; and familiarity with computer use. We defined three groups: two experimental groups, BlindAid and Virtual Cane (Wiimote), and control. All research groups were comparable in age, gender, age of vision loss, and ability to use mobility devices. Each research group was composed of five participants (Table 1). The BlindAid experimental group participants explored the unfamiliar spaces visiting VEs through a Phantom device. The Virtual Cane experimental group participants explored the unfamiliar spaces as VEs with a Wii controller. In contrast, the control group participants investigated the unfamiliar spaces by exploring the RSs. All participants independently explored the unfamiliar RSs. To recruit the participants, we used snowball sampling; each participant was randomly assigned to a research group. Each participant completed an O&M questionnaire to assess O&M abilities. The O&M questionnaire outcomes revealed no differences in O&M ability in any group: in familiar indoor spaces (home or work), none of the participants used a mobility aid; in familiar indoor spaces (small or large shopping mall), 61% of the participants chose to be escorted by a sighted person; in familiar outdoor spaces (their neighborhood with street crossing and public transportation) 60% of the participants used a mobility device (white cane or guide dog); in familiar crowded outdoor and unfamiliar indoor spaces all the participants chose to use a mobility device or to be escorted by a sighted person; in unfamiliar indoor spaces, such as shopping areas, and in unfamiliar outdoor spaces, 88% of the participants chose to be escorted by a sighted person.

### 2.2. Variables

The independent variable in this research was the degree of complexity of spaces explored by the participants (including a simple and a complex space). This level of complexity was related to the size of the space, its structure, and the number of components within it. An O&M rehabilitation specialist examined these spaces to address safety and O&M issues. Three groups of dependent variables were defined: the exploration process, construction of a cognitive map, and performance of orientation tasks in RS. These variables have been defined in our previous research [8,28]. The exploration process included five variables: (1) duration; (2) modes: walk-around mode (exploring the space by walking) or look-around mode (standing in the space in one spot and gathering information as requested about the names of the structure’s components, distance between the structure’s components, objects’ name, and objects’ distance); (3) spatial strategies: random, exploring object area, object-to-object, grid, and perimeter; (4) length of pauses not resulting from a technical issue; and (5) the use of orientation aids in the VE (e.g., teleport action command) or the RS (e.g., using the second hand). Four variables were studied in the building of a cognitive map: (1) components: objects, objects’ location, structural component, and structural components’ location; (2) spatial strategy used for describing the space: starting-point perspective descriptions, items list, object-to-object, or perimeter; (3) spatial representation model used for describing the space: route model, map model, and integrated representation of route and map models; and (4) chronology of the descriptive process. Three variables were related to the orientation tasks performance: (1) the response time of correct orientation task performance (RTC); (2) successful completion of finding the task’s target: arrival at the target (or at the target’s zone), arrival at the target’s zone with verbal assistance, or failure to arrive; and (3) type of path: direct, direct with limited walking around, indirect, or wandering around.

### 2.3. Instrumentation

Three implementation tools and five data collection tools were used. The three implementation tools were:

***Real spaces.*** Two RSs situated on a university campus were used. Two unfamiliar indoor spaces were selected to show how exploration in the VE or RS affected the acquisition of spatial knowledge by people who are blind. The simple space (Figure 1) was a rectangle of 44 square meters containing two windows, five doors, and nine objects (dark green): a communications cabinet (1), an electric cabinet (2), two mailboxes (3–4), a chair (5), a bench (6), a recycling bin (7), and two boxes (8–9). The complex space (Figure 2) was larger and contained two long parallel hallways with two short parallel hallways connecting the long ones; and 12 objects: two benches (1, 7), staircases (2, 16), recycling bin (3), snack machine (4), two round tables (5–6), chair (8), mailbox (9), window (12), electric cabinet (13), pole (14), and box (15).

**The BlindAid.** Development and research of the BlindAid system took place at the MIT Touch Lab as part of a collaborative research project [6,28]. The BlindAid system permits people who are blind to manipulate virtual objects and provides multisensorial feedback. The VE software runs on a personal computer equipped with a haptic device—a Desktop Phantom device (SensAble Technologies, Woburn, MA, USA) and stereo headphones (Sennheiser HD580, Wedemark, Germany) (Figure 3). The Desktop Phantom device provides haptic feedback from the tip of the Phantom, similar to that generated by the tip of a white cane (e.g., stiffness and texture). The haptic device can simulate different degrees of ground texture and stiffness (e.g., marble floor or rubber floor) and the textures and stiffness of different objects (e.g., table or sofa). The system includes surrounding audio feedback that conveys sounds to the users as if they were standing in the VE.

The virtual workspace is a rectangular box that corresponds to the usable physical workspace of the Phantom, and the user avatar is always contained within the workspace. To move the virtual workspace within the VE in order to explore beyond the confines of the workspace, the user presses one of the arrow keys. Each arrow key press shifts the workspace half of its width in the given direction. The VR system includes unique action commands that are available to the user only in this VR system and not in the RS. The six action commands on the computer’s numeric keypad include teleport, pause, start, additional audio information, exploring the VE’s structure layer without the objects in it, and exploring the VE’s structure with the objects. Further technical details about the system were presented in our earlier paper [30]. For this study, eight VEs were designed to train the participants on the use of the BlindAid system (Figure 4). These eight VEs differed in their level of complexity: size, shape, components (structure and objects), and components’ location.

To evaluate the participants’ spatial ability, two additional VEs were designed in the BlindAid system based on the RSs that were chosen earlier: a simple space (Figure 1) and a complex space (Figure 2). These simple and complex VEs are identical to the RSs and to the Virtual Cane VEs in layout and components.

**The Virtual Cane.** The Nintendo Wii controller (Wiimote) system was developed in collaborative research with the team at the Computing and Informatics Research Centre at Nottingham University. The Wii technology is popular, low cost, and easy to use with a standard personal computer (PC) [31] (Figure 5).

The Wiimote includes tracking technologies (e.g., accelerometer and infrared camera); the Wiimote system permits people who are blind to interact with the remote controller (Figure 6) as a white cane and in addition as a handheld camera in the VE. The VE interface was developed by the Windows-based Wii Controller Interface Suite (WiiCi). The WiiCi tools enable a connection between a personal computer and controller [32].

The Wiimote system was connected to a PC and the participants were seated next to it. The participant held the Wiimote remote controller and the Nunchuck (Figure 6) in both hands and wore stereo headphones (Sennheiser HD580). The Wiimote remote controller has five buttons: button “+” produces audio feedback about the object distance; button “−” produces audio feedback about the object name; button “A” toggles from walk-around to look-around mode; the arrows button operates the sonar system; and button “Home” offers an auditory description of the space.

The Virtual Cane system operates in two modes:

Look-around mode. The look-around mode is realized through the movement and orientation of the Wiimote remote controller. The avatar point of view is directly slaved to that of the remote controller. Users receive auditory and tactile (vibration) feedback by using the remote controller for scanning for objects in front of them. Each structure or object component in the VE has unique auditory and tactile feedback. The audio feedback includes the object’s name and its distance. Tactile feedback vibration is activated and differs in accordance with the distance to an object (determined via ray-casting from the point of reference of the virtual Wii controller). A constant rumble is triggered by a collision with an object or the structure’s components. When the user is out of the look-around mode, tactile feedback is given as a constant rumble. Moving the avatar from one area to another produces a whooshing sound. The look-around mode is available only in this orientation VR system, which was designed especially for users who are blind.

The walk-around mode. The walk-around mode is operated by tilting the Nunchuck controller in four directions (left, right, forward, or back). With the tilt direction of the Nunchuck controller, the user avatar moves in a fixed 15-degree turn in the VE. The walking speed is related to the severity of tilt. Auditory feedback (right or left) is heard before the activation of a turn. Furthermore, auditory feedback indicating footsteps is received as the avatar walks in the VE. Further technical details about the Virtual Cane system can be found in our earlier paper [28].

To allow users to train themselves in operating this Virtual Cane system, eight VEs were built (identical to the BlindAid training VEs, Figure 4). Further, simple and complex VEs were designed identical to the RSs (Figure 1 and Figure 2) and BlindAid VEs in layout and components.

The five data collection tools are as follows:

**O&M questionnaire.** The questionnaire is a self-evaluation of O&M skills and abilities. This questionnaire included 50 questions regarding the participant’s O&M abilities in familiar and unfamiliar indoor and outdoor spaces. The O&M questionnaire was the same as employed in previous research [6,8,28,29]. Four O&M rehabilitation specialists assessed the questionnaire.

**Exploration task.** All participants were invited to explore both spaces (simple and complex spaces) in a limited timeframe that was recommended by an O&M rehabilitation specialist (40 min for exploring the simple space and 60 min for exploring the complex space). The control group explored the RSs and both experimental groups explored them via the virtual systems (BlindAid or Virtual Cane).

**Verbal description task.** After the exploration task, the participants were asked to give a verbal description of the space. The verbal description served as an instrument to measure the cognitive map that the participants constructed as a result of their exploration in the RS or VE. The description tasks were video recorded and transcribed.

**RS orientation tasks.** Following the exploration and verbal description tasks, the participants were asked to perform orientation tasks in the RSs (simple and complex): (1) object-oriented tasks—the participants were requested to travel from the original exploration starting point (exploration task) to an object (three tasks in the simple space and two tasks in the complex space); (2) perspective-change tasks—the participants were asked to go from a new starting point to an object (three tasks in the simple space and two tasks in the complex space); and (3) point-to-the-location tasks—the participants were located in the original exploration starting point and were requested to point with their hand at the location of six different structural or object components.

**Observations.** Screen recordings were made of the experimental groups’ activities in the BlindAid or Virtual Cane (Wiimote) systems and videos of the participants were recorded. Synchronization of both video recordings was performed on the researcher’s computer using Camtasia 2 (screen recording software). The control group participants were video recorded in the RS exploration tasks. All research groups were video recorded during their verbal description and RS orientation tasks.

### 2.4. Data Analysis

The data were analyzed employing quantitative and qualitative methodologies. To assess the participants’ O&M performance in the exploration task, verbal description task, and RS orientation tasks, we used coding schemes that were principally developed in previous research and evaluated by four O&M rehabilitation specialists [6,8,9,29]. We developed coding schemes for each task based on the previous coding schemes analyses and the O&M literature [33,34]. Data analysis was performed using Microsoft Excel^®^ and Mangold Interact^®^ software.

### 2.5. Procedure

The research included three to seven sessions: in the first session, all participants signed the consent form and answered the O&M questionnaire. In the second session, all participants individually explored the VEs or RSs. The experimental group participants had a total of six sessions: four sessions were devoted to learning to operate the VE system, and two sessions were spent exploring the simple and complex VEs; the duration of each session was 90 min. The control group participants had two sessions to explore the simple and complex spaces; as for the experimental groups, the duration of each session was 90 min. Following each exploration task (simple and complex space), all participants verbally described the space and then performed the RS orientation tasks. All participants performed all tasks in the same protocol order.

### 2.6. Research Limitation

The limitation of this study to 15 subjects arose from its exploratory nature and the challenges of participant recruitment. The small sample prevented the running of statistically significant tests, but the data collected reveal interesting distinctions that can be further evaluated with a larger sample in the future.

## 3. Results

Research Question 1: How do people who are blind explore unfamiliar spaces using BlindAid or Virtual Cane, or in RS?

Both experimental groups (BlindAid and Virtual Cane (Wiimote) systems) and control group (RS) participants explored the VE or the RS independently. In the simple space, all the participants performed the exploration task in less than the suggested exploration time (40 min). The BlindAid experimental group took an average of 00:19:43 min. The Virtual Cane experimental group took an average of 00:41:44 min. Three participants performed in less than the suggested exploration time (40 min), and the other two participants needed more time. The control group took an average of 00:04:39 min, about seven times less than the suggested exploration time (30 min). The participants in the BlindAid and the control groups were able to explore the spaces only by the walk-around mode (by choosing one of the spatial strategies such as perimeter, object-to-object, and other). In contrast, the Virtual Cane experimental group participants were able to implement both look-around and walk-around modes (Table 2). The BlindAid participants used the walk-around mode in 98% of the exploration time, mainly the perimeter strategy (95%), and used 2% (00:00:24) of their exploration time for pauses. The results showed that the Virtual Cane participants mainly used the look-around mode to explore the VE (74%), walked around in the VE for only 9% of the time, and paused for 17% of the time (00:07:06). Their most used walk-around mode spatial strategy was the object-to-object strategy. The control group participants who explored the RS used the walk-around mode for 94% of their exploration time, by using mostly the perimeter strategy (84%); only 6% (00:00:17) of their exploration time was used for pauses.

To explore the complex VE the BlindAid participants took an average of 00:36:46 min; the Virtual Cane experimental group took an average of 00:53:19 min; and the control group participants took an average of 00:09:14, about four to six times less compared with the BlindAid and Virtual Cane groups (Table 3). To explore the complex space, the BlindAid experimental group mainly used the perimeter strategy (96%), with very few pauses during the exploration (00:00:44; 2%). The Virtual Cane experimental group performed the tasks by using mostly the look-around mode (73%), using less the walk-around mode (12%), and making use of long pauses (15%; 00:08:00). The most-used spatial strategies were the object-to-object strategy (11%) and perimeter (9%). The control group participants explored the RS by walking for 99% of their exploration time, mainly using the perimeter strategy (82%), with pauses taking only 1% (00:00:05) of their exploration time.

A comparison of spatial behavior in the simple and complex spaces shows that there was an almost equal division of exploration time among the three groups. This comparison highlights four main differences: exploration duration, spatial strategies, pauses, and exploration aids. The BlindAid experimental group took four times longer than the control group to explore each space; similar results were found with the second experimental group—using the Virtual Cane took six to nine times longer compared with the control group. The choice of exploration mode and spatial strategies depended upon which space was being explored (BlindAid, Virtual Cane, or RS). The participants in the BlindAid experimental group and control group managed to use only the walk-around mode and they mainly used the perimeter strategy, while the Virtual Cane experimental group mostly chose to use the look-around mode and object-to-object strategy in the walk-around mode, with the perimeter as their secondary strategy. During the exploration of both spaces, the BlindAid (00:00:24; 00:00:44) and control group (00:00:17; 00:00:05) participants tended to use shorter pauses compared to the Virtual Cane participants (00:07:06; 00:08:00).

During the exploration process, a variety of exploration aids (action commands) was used, depending on the VE or the RS. The BlindAid system allows its participants to use three action commands: teleport (move the user’s avatar to the starting point), obtain additional auditory information, and explore the VE with or without objects in it. The Virtual Cane system allows the use of the teleport action command and the look-around and walk-around modes to obtain name or distance information. In the RS the participants walked to the starting point and used their second hand to explore the space.

The results show that 66% of the BlindAid participants used the teleport action command, all the participants used the additional auditory information action command, and only one participant chose to explore the complex VE without objects for the first half of her exploration duration. All the Virtual Cane experimental participants used the teleport action command to move the user’s avatar to the starting point during their exploration. Using look-around mode, the participants asked for four different types of auditory information: object name, object distance, structure component name, and structure component distance. The results showed that, while using the look-around mode, for 40% to 43% of the time (simple vs. complex spaces), the participants requested auditory feedback about the object or structure component’s name, and 34% to 29% (simple vs. complex) of the time they asked for information about the object or structure component’s distance (by pointing with the Wiimote remote controller at virtual components in the VE, an audio feedback was received describing the number of steps between the avatar and the pointed virtual components). Additionally, in a scenario in which the participants collided with a virtual component, they selected distance information rather than its name. Although the Virtual Cane allows the use of the look-around mode to indicate heights, only one participant in the simple space made use of this action command and then only for a few seconds.

The control group participants used their second hand to explore the RS during their exploration time in the simple (52%) or complex (43%) spaces. Only two participants chose to return to the starting point.

Research Question 2: What were the participants’ cognitive mapping characteristics after exploration using the BlindAid or Virtual Cane, or in RS?

After each exploration task, all the participants were asked to verbally describe the space. We evaluated their verbal descriptions with four variables: space components (object, object location, structural component, and structural component location); spatial strategy; spatial representation model; and chronology description.

We examined verbal descriptions of the simple (Table 4) and the complex (Table 5) spaces. In their description of the simple space, the BlindAid experimental participants described an average of 59% of the total components (structure and objects) that were placed in the simple space; the Virtual Cane participants described an average of 69% of the total components, compared to an average of 40% by the control participants. All the research participants included more objects than structural components in their verbal descriptions. The participants used all types of spatial strategies to describe the space (e.g., perimeter, object-to-object, starting point, area, and list). The main difference was found in the spatial representation that was used during the verbal description: the BlindAid experimental group and control group participants mainly used a route model, compared to the Virtual Cane experimental group participants, who employed a map model. Most descriptions began with mention of a structural component, except for one participant from the BlindAid experimental group, who started his description with the content description. 

For the complex space (Table 5), the BlindAid experimental participants mentioned in their description an average of 47% of the total components that were located in the VE and the Virtual Cane participants mentioned an average of 44% of the total components, compared to 30% for the control participants. The participants included more objects than structural components in their verbal descriptions and used all types of spatial strategies to describe the space. In the BlindAid experimental group’s verbal descriptions, four participants employed a route model and one participant listed them. Verbal descriptions by the Virtual Cane experimental group included the use of a map model by two participants, use of a route model by two participants, and list by one participant; similar results were found in the control group, where a map model was used by two participants and a route model by three participants. As they did in the simple space, here, most of the participants began their verbal descriptions with a structural component, except for two participants from the BlindAid experimental group and the control group (each), who first described the objects.

The results show differences among the three research groups and in both spaces. Participants in both experimental groups (in both spaces) included greater detail in their verbal descriptions compared to the control group participants. The BlindAid experimental group and control group described both spaces using mainly a route model, as opposed to two participants from the control group who used the map model in the complex space. In contrast, the Virtual Cane experimental group mainly described the simple space using a map model, while in describing the complex space, two participants constructed a map model.

Compared with the complex space, the simple space was described in more detail by all participants.

Research Question 3: How did the control group and the two experimental groups (BlindAid and Virtual Cane) perform the RS orientation tasks?

To answer the third question, the performance of orientation tasks by participants in the RSs was evaluated. Orientation tasks in the simple space included three object-oriented tasks, three perspective-change tasks, and one point-to-the-location task (Table 6), and in the complex space, two object-oriented tasks, two perspective-change tasks, and a point-to-the-location task (Table 7). These tasks were evaluated by the response time of correct (RTC) orientation task performance, success, and the path that was chosen.

In the simple space, in both experimental groups, participants took four to six times more time to successfully perform the object-oriented tasks (Table 6), and two to five times more time to perform the perspective-change tasks successfully. Success in arriving at the target for object-oriented tasks was similar among the three research groups (73–74%). In perspective-change tasks, 87% of the BlindAid experimental participants, 67% of the Virtual Cane experimental participants, and 73% of the control participants were successful. Regarding the type of path, in the object-oriented tasks, 60–67% of both experimental groups walked directly to the target, compared to 73% for the control group; in perspective-change tasks, 87% of the BlindAid experimental participants, 47% of the Virtual Cane experimental participants, and 73% of the control participants walked directly to the target. In the point-to-the-location task, most of the participants in the three groups were able to point accurately to the target objects (67% of the BlindAid participants, 80% of the Virtual Cane participants, and 73% of the control participants).

In the complex space (Table 7), the experimental group’s participants took two times more time to complete successfully the object-oriented tasks, and two to three times more time to complete the perspective-change tasks successfully. Two of the Virtual Cane experimental participants stayed in the entrance lobby and did not explore the entire VE. These limitations in exploration affected performance in the orientation RS tasks. In their first object-oriented task, they tried to transfer their VE landmarks to the real complex space to ground their spatial knowledge. Following the first object-oriented task, they became more self-assured, as also seen in the task durations, which became shorter even when a target object was farther away, and also in the perspective-change tasks. In performing object-oriented tasks, similar results were found among the BlindAid and control groups in their success in arriving at the target (60%), with 40% success for the Virtual Cane experimental group. Their performance in perspective-change tasks varied, with 80% success by the control group, 60% by the Virtual Cane experimental participants, and 50% of the BlindAid experimental participants in arriving at the target. Regarding the type of path in the object-oriented tasks, 60% of the BlindAid experimental group walked directly to the target, compared to 10% for the Virtual Cane experimental group, and 40% for the control group. In perspective-change tasks, 60% of the control group and 40% of both experimental groups walked directly to the target. In the point-to-the-location task, the BlindAid experimental participants pointed successfully at a rate of 70%, while 38% of the Virtual Cane experimental group succeeded in this task.

In the point-to-the-location task, differences were found among the three research groups. Most of the BlindAid experimental participants (70%) were able to point accurately to the target objects, while 38% of the Virtual Cane participants were able to point accurately; one participant failed as a result of a mirror distortion, and there was only 17% success in the control group.

This research took place in a real campus, which includes sounds that are particular to a crowded university setting. A selection of these sounds occurred in the VE and served as landmarks for the participants, for example, the sound of a snack machine or elevator. The experimental participants used these auditory landmarks eight times compared to only four times by the control participants.

## 4. Discussion

This study followed previous research that examined the use of a multisensorial VR to perceive spatial information in a VE using the Virtual Cane system [28]. The previous research results showed that exploring VEs through the look-around model influenced the spatial ability of the participants to construct a cognitive map based on the map model. To determine the influence of the look-around mode on the participants’ spatial ability, this study compared the spatial ability of two experimental research groups using different VR systems exploring the same spaces. The first experimental group used the BlindAid system, which only allows users to walk in the VE to explore the space (similar to RS), and the Virtual Cane system, which offers users the choice of look-around or walk-around modes to explore the space.

This discussion addresses the research goals, which focus on the impact of exploring multisensorial VE systems or RS on the spatial abilities of people who are blind, and the use of unique user-interface action commands and their influence on the exploration process, the construction of cognitive maps, and the application of this spatial knowledge in orientation in the RS. To address these aims, we designed two Ves, one simple and one complex, which were exact representations of the RSs, for both VR systems. Both spaces were unfamiliar to all research participants.

### 4.1. The Impact of Multisensorial VE Systems on the Spatial Abilities of People Who Are Blind

This research paradigm involved three phases: exploring unfamiliar space, constructing a cognitive map, and performing orientation tasks in the RS [35]. These three phases involved the transformation of spatial abilities and spatial information from the RS to the VE and vice versa. The exploration in the multisensorial Ves and the RS was based on previously acquired spatial skills and strategies, which focused on how to explore and collect spatial landmarks and clues in unfamiliar spaces. Later, these explorations assisted participants in constructing a cognitive map. This spatial information (from the exploration process and cognitive map) was transferred from the VE to the RS during RS orientation tasks. Results drawn from the constructed cognitive maps show that all participants who explored the VEs included more details of the components (structure and objects and their location) compared with the participants who explored only the RSs. In regard to the RS orientation tasks performance, three aspects were of interest: RTC, successful completion of finding the task’s target and point-to-the-location, and type of path. The participants in the experimental groups required more time to perform all the tasks. In regard to their success in finding the task’s target and point-to-the-location, participants who explored the VEs successfully performed most of the RS orientation tasks (50%) or were equally successful (33%) compared with the control group (17%). In choosing the direct path to their target, the results between the experimental groups and control group were equal (50%). These performance-success results demonstrate that exploration through multisensorial VR systems results in spatial ability at a level better or equal to that achieved in RS exploration. In addition, VE explorers who are blind will need a longer time to find their path in the RS, as this will be their first time walking in the RS after exploring the VE only. Over time, with more practice and frequency of use of VE exploration, this length of time might drop. These results highlight the need for such an orientation tool, especially when independent exploration of unfamiliar space is not possible; in this event, multisensorial VE systems can substitute for the RS.

Similar results were found previously in orientation VR system research [3,6,7,10,11,14,15,16,17,19,20,21,28]; all research participants were able to explore unfamiliar space independently.

### 4.2. The Impact of Unique Action Commands Embedded in Multisensorial VR Interface on the Spatial Abilities of People Who Are Blind

Orientation VR systems allow their developers to integrate special action commands that are not available to people who are blind in RS. In this research, we examined if and how VR technology affects spatial and cognitive abilities. The BlindAid and Virtual Cane systems included action commands that are available to people who are blind only in the VR system. The BlindAid system includes action commands such as teleport (move the user to the starting point), additional audio information about the object, exploring the VE’s structure layout without objects in it, or exploring the VE’s structure with objects. The Virtual Cane system included the teleport action command and the ability to explore the VE using look-around mode. These unique capabilities supported the VE participants during the exploration process and in cognitive map construction and later assisted them in orienting themselves in the RS. The teleport action command was used by all VR users by teleporting them directly to the entrance point in an easy, short, and simple way. The look-around mode action command, used only by the Virtual Cane participants, influenced exploration, collection of spatial information, and manner of constructing a cognitive map.

By comparing the spatial tasks of the two experimental groups, we hoped to learn more about the impact of the user-interface orientation components on spatial and cognitive abilities. The Virtual Cane participants used mainly the look-around mode, which affords the user the ability to stand in one location and to scan the components of the space. To explore RS, people who are blind rely on many information units, which they need to collect in order to decide how to navigate the space. In this way, they compensate for the lack of access to distant landmarks, which have been found to be very useful to sighted people [36]. Unlike the RS, the Virtual Cane system provides this access to distant landmarks. It allows the user to collect information about an object’s identity and its distance from the user without the need to walk to it. In addition to being accurate and detailed, this VE exploration created an information load [28]. Using the look-around mode affects the participants’ exploration process (duration, spatial strategy, and pauses) and their cognitive map (spatial representation). In the exploration task, they needed a longer time to explore the VE and much longer pauses, compared with the BlindAid or control groups. These pauses were not technical pauses; rather, they were used to recall the spatial information, to organize, to create the relations between each component (direction and distance), and finally to construct a cognitive map. These spatial processes might affect cognitive load, exploration duration, and pauses.

The Virtual Cane participants used the walk-around mode for a short time, mainly the object-to-object strategy, as a spatial strategy. In contrast, the BlindAid and the control groups, who used only the walk-around mode, mainly used the perimeter strategy. These results mirrored practices found in O&M rehabilitation programs, which recommend the use of the perimeter strategy first, followed by grid or object-to-object strategies, in unfamiliar indoor spaces [37].

The look-around mode had one more effect on the type of spatial representation in the construction of the cognitive map. Most of the Virtual Cane participants used the map model, in contrast to the BlindAid and control group participants, who mainly used the route model or mentioned the components as a list of items. There are two ways to represent space, by using a route model or map model; most sighted people use both models. In O&M rehabilitation training, for safety reasons, the main spatial representation that is learned is the route model [37]. This approach guides the person who is blind to concentrate on his or her target path without paying attention to other components in the space that are not related to the target path; however, situations arise in which the map model is more efficient, for example, when the path is blocked and there is a need to find an alternate path.

The unique action command of the BlindAid system focuses on exploring the space in layers: the choice of exploring the VE’s structure without the objects it contains or with the objects. This action command is not available for people who are blind in the RS and is not learned in an O&M rehabilitation program, but it does exist in a visual map (paper or digital). This ability to choose the layouts and the degree of spatial resolution is mainly important when we want to adjust the O&M VR system to the user. Only one participant explored the complex VE using this action command. This unique action command might especially aid people who are blind who use a guide dog. They mainly need to collect information about the structure of the space; the guide dog will direct them around obstacles in the form of objects. In contrast, people who are blind who walk with a white cane need to survey both structure and object components to be able to avoid colliding with objects. The ability to adapt the VE to the participant’s needs and to his or her mobility aid is a valuable component of the VR user-interface approach. To integrate this widely, it should be integrated into O&M rehabilitation training.

### 4.3. Implications for Researchers and Developers

Further research should explore the integration of VR systems in O&M rehabilitation programs and the impact of the special action commands on the spatial ability of people who are blind. For example, it should evaluate the impact of the VE exploration using the look-around mode on RS orientation tasks that encompass the need to find an alternate path. It would also be useful to study differences in spatial orientation behavior of participants who use a guide dog versus a white cane and the implications of these differences for the design of a VE user interface.

Further development of the next orientation VR systems should be address three factors: intuitive and simple interfaces [25], O&M rehabilitation program theories [37], and adaptive action commands that are able to enhance spatial abilities in a manner not available in RS as discussed above.

Finally, future VR systems might be based on smartphones [38,39,40], wearable technologies [41,42], and crowd-sourced navigation technologies. Since smartphones, as mainstream devices, are more affordable and easier to use, most people who are blind and O&M specialists use them. For example, the X-Road project [38] uses a smartphone and a headset to allow users who are visually impaired to explore RS via a VR system. A crowd-sourced navigation app powered and monitored by the walking community, such as the Waze application, can aid people who are blind to update spatial information about their target path or area.

## Figures and Tables

**Figure 1 sensors-22-01307-f001:**
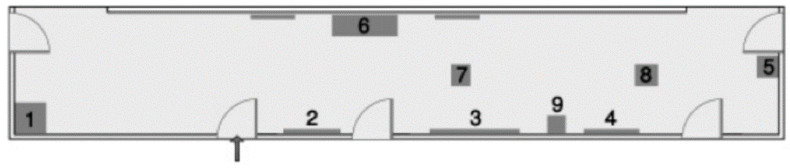
Simple space.

**Figure 2 sensors-22-01307-f002:**
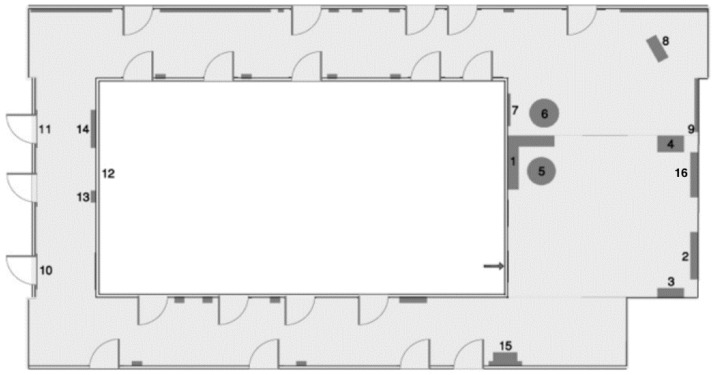
Complex space.

**Figure 3 sensors-22-01307-f003:**
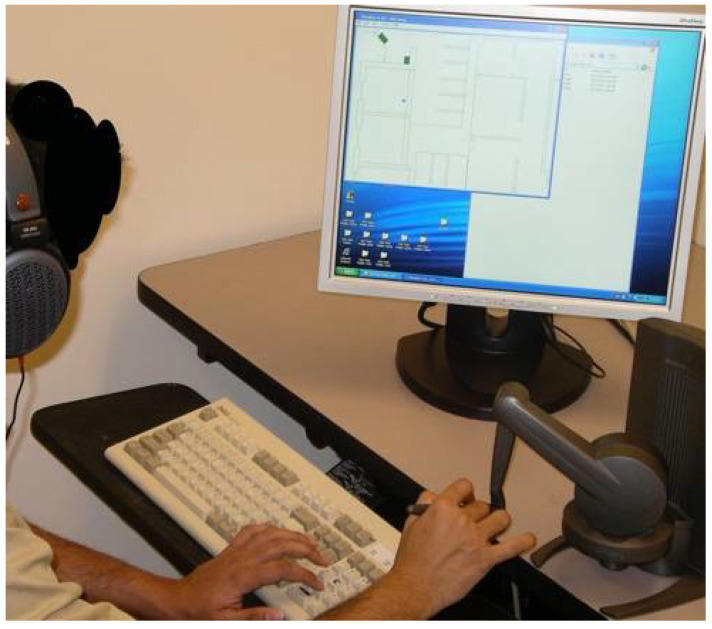
BlindAid system.

**Figure 4 sensors-22-01307-f004:**
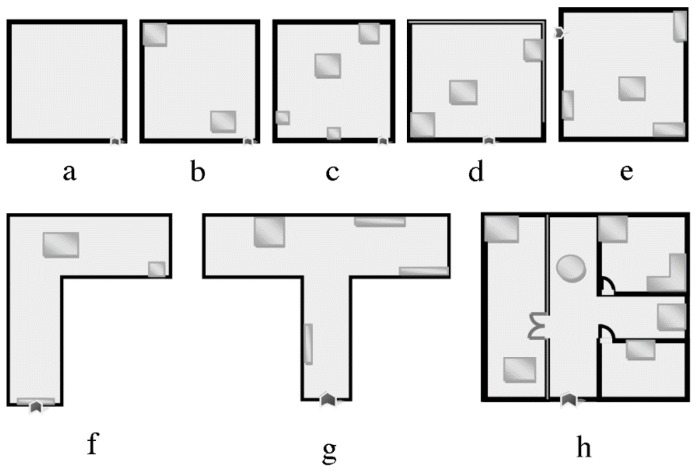
VE training spaces: (**a**) an empty squre room; (**b**) a squre room with one object attached to wall, and second located far from the wall; (**c)** a squre room with three objects in different sizes attached to the wall, and one more located far from the wall; (**d**) a rectangle room with two objects in different sizes attached to the wall, and one more located in rooms’ center; (**e**) a rectangle room with three objects attached to the wall, and one more located in rooms’ center; (**f**) a “L” shape space with three objects: a door, one object attached to the wall, and one more located in spaces’ center; (**g**) a “T” shape space with five objects: a door and four objects in different sizes attached to the wall; and (**h**) a squre space devided to three rooms with a corridoe in “T” shape with a door and seven objects in different sizes and locations.

**Figure 5 sensors-22-01307-f005:**
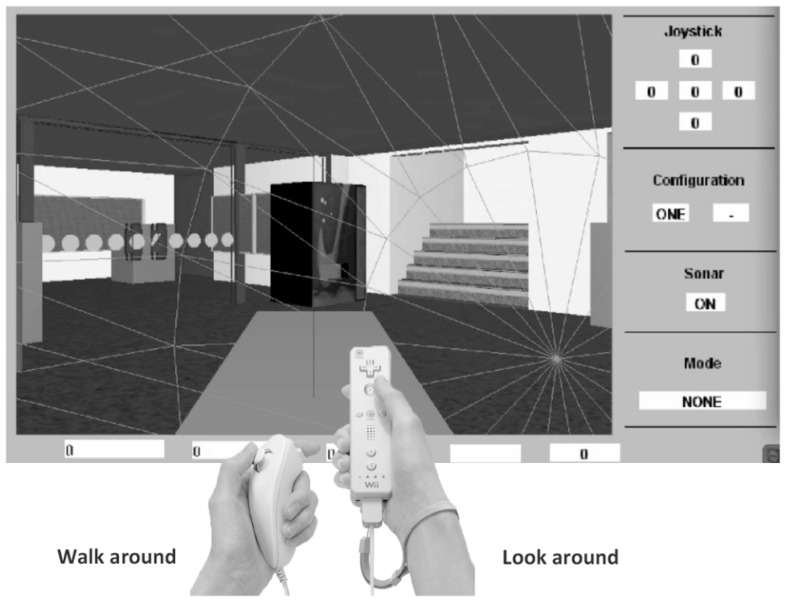
Virtual Cane system.

**Figure 6 sensors-22-01307-f006:**
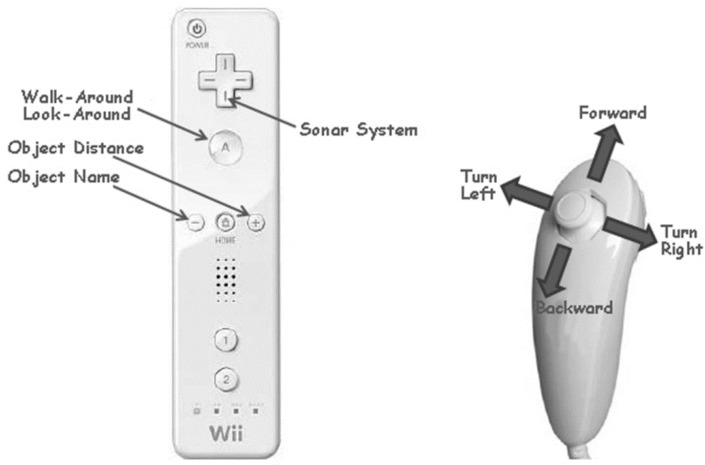
The Wiimote remote controller (**left**) and Nunchuck controller (**right**).

**Table 1 sensors-22-01307-t001:** Research participants.

	Age Mean	Gender		Age Of Vision Loss		O&M Aids	
		Female	Male	Congenitally	Adventitiously	White Cane	Guide Dog
BlindAid experimental group (*n* = 5)	43 (28–59)	2	3	5	0	4	1
Virtual Cane experimental group (*n* = 5)	30 (25–40)	3	2	3	2	3	2
Control group (*n* = 5)	40 (27–56)	1	4	2	3	3	2

**Table 2 sensors-22-01307-t002:** Exploration process in simple space.

	*n*	Duration (min)	Look-Around Mode		Walk-Around Mode Spatial Strategy			Pauses (min; %)	Second Hand
			Name	Distance	Perimeter	Object-to-Object	Other		
BlindAid experimental group	1	31:00	NA	NA	100%	0%	0%	0%	NA
	2	24:30	NA	NA	99%	0%	0%	0%	NA
	3	4:56	NA	NA	98%	0%	1%	0%	NA
	4	28:02	NA	NA	90%	0%	0%	02:31; 9%	NA
	5	10:08	NA	NA	86%	0%	12%	00:06; 1%	NA
	Mean	19:43	NA		95%	0%	3%	00:24; 2%	NA
Virtual Cane experimental group	1	29:24	37%	33%	2%	6%	0%	06:46; 23%	NA
	2	49:39	28%	54%	5%	6%	0%	03:29; 7%	NA
	3	38:32	41%	33%	0%	3%	0%	08:29; 22%	NA
	4	58:45	45%	21%	4%	3%	0%	16:27; 28%	NA
	5	32:18	50%	28%	6%	8%	0%	02:16; 7%	NA
	Mean	41:44	74%		3%	5%	0%	07:06; 17%	NA
Control group	1	3:44	NA	NA	87%	0%	3%	00:22; 10%	90%
	2	11:27	NA	NA	62%	2%	29%	00:48; 7%	97%
	3	2:01	NA	NA	81%	0%	6%	00:16; 13%	72%
	4	1:26	NA	NA	100%	0%	0%	0%	7%
	5	4:38	NA	NA	89%	11%	0%	0%	63%
	Mean	4:39	NA		84%	3%	8%	00:17; 6%	52%

**Table 3 sensors-22-01307-t003:** Exploration process in complex space.

	*n*	Duration (min)	Look-Around Mode		Walk-Around Mode Spatial Strategy			Pauses (min; %)	Second Hand
			Name	Distance	Perimeter	Object-to-Object	Other		
BlindAid experimental group	1	22:45	NA	NA	90%	1%	0%	01:36; 7%	NA
	2	73:04	NA	NA	99%	0%	0%	0%	NA
	3	13:16	NA	NA	100%	0%	0%	0%	NA
	4	55:34	NA	NA	92%	0%	3%	01:07; 2%	NA
	5	19:10	NA	NA	100%	0%	0%	0%	NA
	Mean	36:46	NA		96%	0%	1%	00:44; 2%	NA
Virtual Cane experimental group	1	26:36	51%	22%	1%	7%	0%	05:35; 21%	NA
	2	63:11	26%	49%	3%	13%	0%	03:47; 6%	NA
	3	44:43	46%	32%	2%	2%	0%	07:09; 16%	NA
	4	86:53	43%	23%	5%	6%	0%	19:07; 22%	NA
	5	45:13	51%	20%	9%	11%	0%	03:10; 7%	NA
	Mean	53:19	73%		4%	8%	0%	08:00; 15%	NA
Control group	1	7:18	NA	NA	78%	6%	15%	0%	76%
	2	19:23	NA	NA	82%	0%	18%	0%	100%
	3	10:05	NA	NA	84%	3%	9%	0%	4%
	4	3:01	NA	NA	100%	0%	0%	0%	4%
	5	6:24	NA	NA	68%	0%	23%	00:35; 9%	60%
	Mean	9:14	NA		82%	2%	13%	00:05; 1%	43%

**Table 4 sensors-22-01307-t004:** Verbal description process of simple space.

	*n*	Space Components	Spatial Strategy	Spatial Representation	Chronology
BlindAid experimental group	1	65%	Perimeter	Route model	Structure
	2	87%	Perimeter	Route model	Structure
	3	26%	List	List	Content
	4	78%	Perimeter	Route model	Structure
	5	37%	Area	Route model	Structure
Mean		59%		4 Route model; 1 List	4 Structure; 1 Content
Virtual Cane experimental group	1	66%	List	Map model	Structure
	2	79%	Starting point	Map model	Structure
	3	68%	Perimeter	Route model	Structure
	4	72%	Perimeter & object to object	Map model	Structure
	5	62%	Starting point & object to object	Map model	Structure
Mean		69%		4 Map model; 1 Route model	5 Structure
Control group	1	36%	Perimeter	Route model	Structure
	2	54%	Area	Map model	Structure
	3	27%	Starting point	Route model	Structure
	4	15%	List	List	Structure
	5	70%	Starting point	Route model	Structure
Mean		40%		3 Route model; 1 Map model; 1 List	5 Structure

**Table 5 sensors-22-01307-t005:** Verbal description process of complex space.

	*n*	Space Components	Spatial Strategy	Spatial Representation	Chronology
BlindAid experimental group	1	42%	Perimeter & list	Route model	Structure
	2	63%	Perimeter	Route model	Structure
	3	50%	Perimeter	Route model	Structure
	4	53%	Perimeter & area	Route model	Content
	5	28%	Area	List	Structure
Mean		47%		4 Route model; 1 List	4 Structure; 1 Content
Virtual Cane experimental group	1	20%	List	List	Structure
	2	46%	Area	Route model	Structure
	3	41%	Perimeter	Route model	Structure
	4	53%	Perimeter & object to object	Map model	Structure
	5	59%	Area	Map model	Structure
Mean		44%		2 Map model; 2 Route model; 1 List	5 Structure
Control group	1	30%	Starting point	Map model	Structure
	2	50%	Area	Map model	Structure
	3	30%	Object to object	Route model	Structure
	4	14%	Perimeter & starting point	Route model	Content
	5	27%	Perimeter & starting point	Route model	Structure
Mean		30%		3 Route model; 2 Map model	4 Structure; 1 Content

**Table 6 sensors-22-01307-t006:** Success in orientation tasks in simple space.

	*n*	Object-Oriented			Perspective-Change			Point-to-the-Location
		RTC (s)	Success	Direct Path	RTC (s)	Success	Direct Path	
BlindAid experimental group	1	9	67%	67%	37	100%	100%	83%
	2	18	100%	100%	53	67%	67%	33%
	3	26	100%	100%	25	100%	100%	100%
	4	194	33%	0%	92	67%	67%	50%
	5	NA	67%	33%	NA	100%	100%	67%
	Mean	62	73%	60%	52	87%	87%	67%
Virtual Cane experimental group	1	57	100%	100%	192	100%	67%	83%
	2	24	67%	67%	129	33%	0%	67%
	3	34	67%	67%	68	67%	67%	83%
	4	39	67%	67%	74	67%	33%	67%
	5	68	67%	33%	93	67%	67%	100%
	Mean	44	74%	67%	111	67%	47%	80%
Control group	1	7	33%	33%	27	33%	33%	67%
	2	10	100%	100%	27	100%	100%	100%
	3	17	100%	100%	9	100%	100%	100%
	4	9	100%	100%	21	67%	67%	83%
	5	10	33%	33%	22	67%	67%	17%
	Mean	11	73%	73%	21	73%	73%	73%

**Table 7 sensors-22-01307-t007:** Success in orientation tasks in complex space.

	*n*	Object-Oriented			Perspective-Change			Point-to-the-Location
		RTC (s)	Success	Direct Path	RTC (s)	Success	Direct Path	
BlindAid experimental group	1	NA	0%	0%	NA	50%	50%	67%
	2	NA	100%	100%	NA	0%	0%	33%
	3	122	50%	50%	25	100%	100%	100%
	4	43	50%	50%	0	0%	0%	67%
	5	237	100%	100%	185	100%	50%	83%
	Mean	134	60%	60%	105	50%	40%	70%
Virtual Cane experimental group	1	0	0%	0%	333	100%	100%	33%
	2	0	0%	0%	132	50%	0%	17%
	3	194	50%	0%	117	50%	50%	50%
	4	185	100%	50%	320	100%	50%	0%
	5	185	50%	0%	0	0%	0%	50%
	Mean	188	40%	10%	180	60%	40%	38%
Control group	1	67	50%	0%	72	50%	50%	33%
	2	44	100%	100%	69	100%	50%	100%
	3	19	50%	50%	42	100%	100%	67%
	4	175	100%	50%	60	100%	100%	50%
	5	0	0%	0%	95	50%	0%	17%
	Mean	76	60%	40%	68	80%	60%	53%

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
