# Peer review of "Virtual Reality Systems as an Orientation Aid for People Who Are Blind to Acquire New Spatial Information"

_sensors, 2022, doi:10.3390/s22041307_

Round 1

Reviewer 1 Report

This work studied the user behaviors in VR and VE scenarios while driven by O&M tasks. A number of tests have been conducted to support the conclusion of the work. The topic is interesting, but the ways of illustrating the work and structuring the paper require extensive revision.

The concerns are stated as follows,

The statement ‘The virtual systems included unique features not available in the real space’ is not readable. What is the “Unique feature” referring to?

Line 15-16, ‘All used the restart command action, one BlindAid participant 15 used the structure layout to explore the space’.

What do you mean?

Line 74, avoid using ‘some’ in scientific papers.

Line 113-120, the goal and research questions are weak in terms of scientific contribution and technical soundness. Experimental trials are not solid enough for rigorous research works.

Table 2, difficult to read.

line 244, the authors mentioned ‘two VEs were designed based on the RSs’. However, in Figure 4, seven VE training spaces are shown. Why? Please clarify.

The statement ‘Both VEs are identical to the RSs and to the Virtual Cane VEs in layout and components’ requires further explanation. In my opinion, it is not the same thing in Figure 1, Figure 2, and Figure 4. Which parts exactly of these diagrams refer to? Please clarify.

The statement ‘participants in the experimental groups successfully performed most of the RS orientation tasks or were equally successful compared with the control group’ seems to be inconsistent with the previous data results. In the previous data, there are many data that the control group is superior to those of the experimental group. In the meantime, there are cases that the control group that takes less time. See line 539 and line 554. Please explain.

Line 640-641, please revise.

Line 666-667, please revise.

Well, most of the references are out of date. It is NOT quite acceptable.

Reviewer 2 Report

To examine the impact of virtual environments interface features on the exploration process, construction of cognitive maps, and performance of orientation tasks in real spaces by users who are blind, this study compared interaction with identical spaces using different systems: BlindAid, Virtual Cane, and real space. 
This research is very significant and the results are very interesting in spite of the small number of participants per group. The paper is well organized.

To improve your manuscript, I recommend adding a summary chapter that describes the answers to the following research questions. 
----------------------------
1. How do people who are blind explore unfamiliar spaces using BlindAid or Virtual Cane, or in RS? 
2. What were the participants’ cognitive mapping characteristics after exploration using the BlindAid or Virtual Cane, or in RS? 
3. How did the control group and the two experimental groups
 (BlindAid and Virtual Cane) perform the RS orientation tasks? 
----------------------------  

In Table 2, there is some overlapping of text, so correct it.
Are the sentences in lines 640 and 666 incomplete? 

Round 2

Reviewer 1 Report

Most of the review comments were properly answered. Obviously, the manuscript has been greatly improved after revision, and its readability and scientific rationality are more in line with the needs of readers and the research community.

Therefore, I suggest to positively consider the publication of this work in the journal.